# *Streptococcus mitis* as a New Emerging Pathogen in Pediatric Age: Case Report and Systematic Review

**DOI:** 10.3390/antibiotics12071222

**Published:** 2023-07-23

**Authors:** Claudia Colomba, Valeria Garbo, Giovanni Boncori, Chiara Albano, Sara Bagarello, Anna Condemi, Salvatore Giordano, Laura A. Canduscio, Cristina Gallo, Gaspare Parrinello, Antonio Cascio

**Affiliations:** 1Department of Health Promotion, Maternal and Infant Care, Internal Medicine and Medical Specialties, University of Palermo, 90100 Palermo, Italy; claudia.colomba@libero.it (C.C.); vali.garbo@gmail.com (V.G.); boncori.giovanni@yahoo.it (G.B.); sbagarello@gmail.com (S.B.); annacondemi96@gmail.com (A.C.); gaspare.parrinello@unipa.it (G.P.); antonio.cascio03@unipa.it (A.C.); 2Division of Pediatric Infectious Diseases, “G. Di Cristina” Hospital, ARNAS Civico Di Cristina Benfratelli, 90100 Palermo, Italy; giordano.s@tiscali.it (S.G.); laura.canduscio@libero.it (L.A.C.); 3Division of Radiology, “G. Di Cristina” Hospital, ARNAS Civico Di Cristina Benfratelli, 90100 Palermo, Italy; cristina.gallo@arnascivico.it; 4Infectious and Tropical Diseases Unit, AOU Policlinico “P. Giaccone”, 90100 Palermo, Italy

**Keywords:** VGS, *S. mitis*, pediatric, invasive disease

## Abstract

*Streptococcus mitis*, a normal inhabitant of the oral cavity, is a member of Viridans Group Streptococci (VGS). Generally recognized as a causative agent of invasive diseases in immunocompromised patients, *S. mitis* is considered to have low pathogenic potential in immunocompetent individuals. We present a rare case of sinusitis complicated by meningitis and cerebral sino-venous thrombosis (CSVT) caused by *S. mitis* in a previously healthy 12-year-old boy with poor oral health status. With the aim of understanding the real pathogenic role of this microorganism, an extensive review of the literature about invasive diseases due to *S. mitis* in pediatric patients was performed. Our data define the critical role of this microorganism in invasive infections, especially in immunocompetent children and in the presence of apparently harmful conditions such as sinusitis and caries. Attention should be paid to the choice of therapy because of VGS’s emerging antimicrobial resistance patterns.

## 1. Introduction

*Streptococcus mitis*, an important member of VGS [1], is found as part of the normal microbiota of the human skin as well as of the oropharynx, gastrointestinal, and female genital tracts [2]. Although generally considered to have a low pathogenic potential in immunocompetent individuals, VGS can cause invasive diseases such as bloodstream infections, pneumonia, endocarditis, enteritis, and meningitis in patients with immunocompromised status or other risk factors [3,4]. In the literature pediatric cases of meningitis caused by *S. mitis* are mainly described in patients with leukemia, lymphoma, or neutropenia, while in healthy children, meningitis and other severe conditions caused by *S. mitis* are considered rare [5,6].

In this report, we present a rare case of sinusitis complicated by meningitis and CSVT caused by *S. mitis* in a previously healthy 12-year-old boy. Starting from our case, a systematic review of the literature was performed with the aim of providing the first overview of serious infections caused by this generally saprophytic bacterium in the pediatric population (including both immunocompromised and immunocompetent individuals). Pathogenesis, clinical features, predisposing factors, therapy, and outcomes were analyzed for each case.

## 2. Case Report

A previously healthy 12-year-old boy was admitted to our hospital after seven days of fever, headache, and a recent onset of vomiting. At the time of admission, the patient was alert with a severe headache, diplopia, photophobia, and difficulty maintaining an upright position. A physical examination revealed a fever up to 38.5 °C, normal cardiorespiratory activity, and a painless, treatable abdomen. A mild nuchal rigidity, with pain on passive mobilization and lateral twisting of the neck, was observed in the absence of both Binda and Brudzinski signs. An intraoral examination showed poor oral health status with multiple destructive caries in the upper arch (Figure 1).

Laboratory findings revealed mild anemia and neutrophilia (Table 1). Blood cultures were negative.

A CT-scan showed an enlargement of the sub-tentorial ventricular system associated with sinusitis of the sphenoid, frontal, and maxillary right sinuses.

After a funduscopic examination, a lumbar puncture was performed for suspected meningitis, showing a moderately cloudy appearance of the cerebrospinal fluid with 640 white blood cells (90% polymorphonuclear), a total protein content of 0.3 g/L, and glucose of 57 mg/dL. Microbiological analyses of the CSF were negative.

In the suspect of culture-negative bacterial meningitis, a broad-spectrum empirical antibiotic therapy with ceftriaxone 50 mg/kg/12 h, vancomycin 20 mg/kg/8 h, and metronidazole 10 mg/kg/8 h was started in association with steroid therapy (dexamethasone 4 mg/6 h).

Following a few days of treatment, the patient no longer presented with fever, neck rigidity, headache, or diplopia.

A subsequently performed Contrast MRI showed radiological findings of sinusitis of both sphenoid sinuses, the right maxillary and frontal sinus, and ipsilateral ethmoidal cells, as well as meningeal inflammation and partial thrombosis of the internal jugular veins. After administration of contrast medium, a filling defect of the cavernous sinuses was also reported (Figure 2).

Therefore, anticoagulant therapy with enoxaparin was started.

To exclude the cardiac localization of infection, Echocardiography and color Doppler were performed with negative results.

Orthopantomography showed apical dental granulomas and caries of the upper teeth.

After the ENT consultation, functional endoscopic sinus surgery was performed, and a moderate amount of purulent exudate was drained. The culture of the exudate was positive for *S. mitis,* resistant to amoxicillin, penicillin, and cefuroxime.

Based on the antibiogram, vancomycin administration was discontinued while ceftriaxone and metronidazole were maintained.

Genetic causes of predisposition to thrombosis, as well as any possible cause of immunodeficiency, were ruled out by performing an HIV test, a study of lymphocyte subpopulations, and a hematological evaluation.

On the twentieth day of hospitalization, a second MRI showed a partial regression of the sinusitis and a general improvement of the previously reported brain lesions. It also revealed filling defects in both the internal jugular veins and the right maxillary sinusitis (Figure 3).

The boy was finally discharged in good clinical condition after 37 days of hospitalization. A control MRI performed two months after the discharge showed an almost complete resolution of the thrombosis previously described and a noticeable reduction in the thickening of the mucous membrane of the paranasal sinuses (Figure 4).

The boy is currently in outpatient follow-up.

## 3. Material and Methods

A systematic review of the literature was performed on PubMed and Scopus electronic databases by submitting the query (mitis AND ((baby[Title/Abstract]) OR (child*[Title/Abstract]) OR (pediatr*[Title/Abstract]) OR (paediatr*[Title/Abstract])), with the aim of identifying all cases of invasive disease caused by *S. mitis* in pediatric age reported by 31 October 2022. No filters or language restrictions were applied to the results. Furthermore, all of the listed references were hand-searched, and a citation tracker was used to identify any other relevant papers. All articles involving patients aged 0–18 years with invasive diseases requiring hospitalization (such as bloodstream infection or endocarditis), in which the only identified pathogen was *S. mitis,* were considered eligible for inclusion in our review. Papers without the full text available or lacking adequate information were excluded.

The selected articles were reviewed by two independent authors and judged on their relevance to the subject of the study. The following data were evaluated for each case: age, sex, comorbidities, risk factors, disease associated with *S. mitis* infection, source of isolation of the organism, therapy, and outcome.

This systematic review was performed in accordance with the PRISMA protocol (Reporting Items for Systematic Reviews and Meta-Analyses) [7] after systematic review registration at PROSPERO (the systematic review registration statement code is CRD42022366226).

## 4. Results

A total of 618 potentially eligible articles were found and screened. A total of 542 articles were excluded because they were not inherent with the topic of our research; 15 more were excluded because they concerned adult patients; and 38 could not be examined due to the inaccessibility of the complete article and/or lack of adequate information.

A total of 23 studies were selected for inclusion in the systematic review [2,4,5,6,8,9,10,11,12,13,14,15,16,17,18,19,20,21,22,23,24,25,26], reporting 94 pediatric cases of invasive disease caused by *S. mitis*. Most of the articles were single case reports, while seven were case series. A flow diagram illustrating this selection process is presented in Figure 5.

Clinical features, risk factors, diagnosis, therapy, and outcome of 95 patients (including our case) are reported in Table 2.

The most frequent invasive conditions due to *S. mitis* identified by our review are represented in the graphic below (Figure 6).

Out of the 56 patients whose gender was indicated, 33 (59%) were male and 23 (41%) were female.

The mean age was six and a half years (range: 2 days to 18 years). Eight children were under one year old; the youngest ones were two girls who became ill two days after a physiological birth [17,25].

A total of 78 of 95 (82%) patients were immunocompromised. Out of them, 56 children had a hematological disease, while 22 had an unspecified oncological condition [12].

Hence, approximately 18% of the patients had not a known immunocompromised status, but they still developed a serious condition related to a microorganism normally considered a saprophyte with low pathogenic potential. Among these children, nine had clinical features potentially predisposing to invasive infection:

(1) a 14-years-old girl with Gorham-Stout syndrome with osteolysis skull base and CSF leak had *S. mitis* meningitis [4];

(2–3) an eight-years-old [21] and a one-year-old [23] girl with a ventricular septal defect were diagnosed with endocarditis;

(4) a nine-day-old baby born with hydrocephalus, and thus requiring the placement of a ventriculo-peritoneal shunt a few hours after birth, had meningitis [26];

(5–9) Five children, mean age nine and a halfyears, with chronic heart disease were diagnosed with endocarditis due to *S. mitis* [27].

A total of eight of 95 patients, including our case, were not immunocompromised and had no apparent risk factors.

The outcome was favorable in 88/95 cases. Out of seven deaths, two were due to other causes: a 15-year-old boy with acute myeloid leukemia (AML) died of a contemporary invasive pulmonary aspergillosis [15], and a 17-years-old patient with acute lymphatic leukemia (ALL) died of the underlying disease [19].

## 5. Discussion

*S. mitis* is included in the Viridans Streptococci Group, and in particular in the Mitis group, together with other saprophytes streptococci such as *S. gordonii*, *S. oralis*, and about 20 other related species, normal inhabitants of the oral cavity, respiratory tract, gastrointestinal tract, and human skin [27,28].

Despite their traditionally low pathogenicity, these bacteria are actually identified among the most common causes of bacteremia, as well as toxic shock syndrome, pneumonia, abscesses, endocarditis, and meningitis in both children and adults with immunocompromised status [19,29,30,31,32,33].

Among VGS species, *S. mitis* is the dominant commensal of the oral cavity, and compared to other VGS species, it often causes clinically serious infections in immunocompromised individuals [3], suggesting that *S. mitis* strains have inherently virulent properties.

The genetic analysis performed by Shelburne et al. [3] made it possible to identify a close genetic correlation between *S. mitis* and *S. pneumoniae*, one of the most frequent microbial killers worldwide. Other studies have shown that some genes encoding virulence factors of *S. pneumoniae* are part of the genome of certain strains of *S. mitis* [34,35] and that some representatives of the Mitis group may be mistakenly identified as *S. pneumoniae* [36,37,38,39,40,41,42]. This phenomenon is due to the common evolutionary origin of these organisms as well as the homologous recombination and horizontal gene transfer mechanisms between streptococcal species residing in the same ecological niche [43].

Among others virulence factors, for example, *S. pneumoniae* possess choline containing teichoic acids, which are the anchor structure of choline binding proteins (CBPs). CBPs have an important function in murein metabolism and host- pathogen interactions [44], and they have been occasionally described in some isolates of *S. mitis* [44,45,46].

Such a “genetic closeness” could be responsible for the progressive increasing in virulence showed by these bacteria [47,48,49].

According to our analysis, invasive infections caused by *S. mitis* concern, as expected, predominantly immunocompromised patients with neutropenia. However, clinically severe disease also occurs in a significant number of children with competent immune systems, representing about 20% of the published pediatric cases.

As reported in the results, 8 cases concerning serious invasive infections caused by *S. mitis* in previously healthy children with no immunocompromised status nor evident risk factors, are described in literature.

In our case, the only potential predisposing factors can be identified in the poor oral hygiene of the patient and in the presence of some caries, being *S. mitis* described as a major pathogen involved in the destruction of childhood dental enamel [50,51,52,53]. The presence of destructive caries associated with apical granulomas of the roots of the upper dental arch might have presumably allowed *S. mitis* to invade the paranasal sinuses, causing a severe sinusitis with intracranial complications. Comparable to our case, Yiş et al. [22] reported two cases of meningitis in a previously healthy six-years-old boy and eight-years-old girl, in which the respective sources of infection were probably poor oral hygiene with multiple caries and inflammation of the maxillary sinus.

To date, the overall incidence of neurological complications of sinusitis, as well as their outcomes, have greatly improved thanks to the wide availability of antibiotics, although the mortality rate is still reported to oscillate between 5% and 27% [28], while long-term neurological morbidities such as hemiparesis, aphasia, and epilepsy occur in 13%-35% of survivors [54,55].

Our patient was 12-years-old, confirming the trend for which risks of complications from suppurative sinusitis would increase in pediatric patients above the age of six because of certain anatomical conditions [56,57,58].

The most common complications of sinusitis are orbital and include preseptal or periorbital cellulitis, subperiosteal abscess and optic neuritis [59].

Less frequent intracranial complications include meningitis, epidural abscesses, subdural empyema, intracerebral abscesses and cavernous or sagittal sinus thrombosis [60].

The clinical characteristics of our patient almost immediately suggested the diagnosis of meningitis, presenting fever, vomiting, neck rigidity and visual disturbances.

While among adults meningitis caused by *S. mitis* mostly occurs after invasive procedures like neurosurgical interventions or spinal anesthesia [61,62,63], our review shows that in children the predisposing conditions to *S. mitis*-related meningitis are the early age [16], the immunodeficiency and underlying pathologies which can provide a potential portal for systemic blood infection (such as Gorham-Stout syndrome with loss of CSF [4] and hydrocephalus with ventriculoperitoneal shunt [24]). In addition, as our case suggests, poor oral hygiene or infection of the paranasal sinuses appear to be risk conditions for developing intracranial complications caused by this bacterium. In one case, early-onset meningitis was described in a full-term baby with no obvious predisposing factors [25].

Cerebral white matter involvement during meningitis caused by *S. mitis* has been rarely described. In the two cases of pediatric meningitis reported by Yiş et al. [2,22], brain swelling, diffuse periventricular and white matter hyperintensity, on T2 FLAIR sequences were documented on brain MRI. In the case of our little patient, the MRI showed, in addition to sinusitis, multiple areas of hyperintensity on FLAIR sequences in the cerebellar and cortical areas. Furthermore, the MRI showed a filling defect in both jugular veins, cavernous sinuses, and the right sigmoid sinus, thus suggesting partial venous thrombosis.

CSVT is defined by thrombosis of the superficial (cortical veins, superior sagittal sinus, transverse sinus, sigmoid sinus, and jugular vein) or deep (inferior sagittal sinus, internal cerebral veins, vein of Galen, straight sinus) venous system, and it is an extremely rare complication of upper respiratory tract infections [64]. It is associated with high mortality and morbidity, and it is mostly described in older children, adolescents, and young adults. Severe long-term sequelae are reported in up to 48% of children [65].

No cases of *S. mitis*-related sinusitis complicated by meningitis or CSVT have been previously reported in the literature.

Early and adequate antibiotic therapy is essential in the management of invasive diseases caused by *S. mitis*.

In the case of our patient, empiric antibiotic therapy with vancomycin and ceftriaxone was administered as recommended by the guidelines for the management of bacterial meningitis in children [66,67,68]. All patients diagnosed with *S. mitis*-related meningitis described in the literature received early empiric antibiotic therapy. The most commonly used regimens involved the association of a beta-lactam antibiotic (penicillin, ticarcillin, nafcillin, or ampicillin) with an aminoglycoside (gentamicin, tobramycin, or netilmycin) or the association of a cephalosporin with vancomycin. There were no significant prognostic differences between the two empiric antibiotic regimens. Therapy was subsequently adjusted to the susceptibility of cerebral fluid cultures. The treatment duration was 2–3 weeks for most of the cases. Only one case of death has been reported in a six-year-old girl with ALL [6], while two patients reported sequelae, and one child required a tracheostomy for long-term mechanical ventilation.

A significant reduction in the antimicrobial susceptibility of VGS has been observed over the last few years [69], especially in the pediatric population [70], where Penicillin derivates are often used in clinical practices [71] and prophylaxis [72,73]. Our patient recovered without sequelae, despite the fact that the *S. mitis* isolated was resistant to amoxicillin, penicillin, and cefuroxime. The antimicrobial susceptibility of these bacteria, besides their ability to cause subtle life-threatening diseases, should be carefully monitored because, as stated before, they appear to be able to both acquire resistance genes from neighboring oral microbes and donate resistance genes to more pathogenic streptococci [74,75]. The existence of these “more virulent” bacterial strains could explain how, in some cases, these bacteria are able to cause serious diseases not only in immunocompromised patients but also in healthy patients.

We find it important to shed light on *S. mitis* as a new emerging pathogen in pediatrics. Their probably underestimated role in invasive infections, even in immunocompetent children, could be complicated by their antimicrobial resistance patterns, which justifies the need for more specific and widely shared knowledge.

Furthermore, besides the known risk factors, more attention should be given to apparently less harmful conditions such as sinusitis and neglected oral care, which become relevant for invasive *S. mitis* infections in children. Good oral hygiene, optimal management of sinusitis, and careful clinical evaluation of a child with nonspecific symptoms who has recently undergone dental procedures, even after antibiotic prophylaxis, can thus prevent the spread of *S. mitis* and the consequential development of life-threatening diseases.

Therefore, being aware of the potential antimicrobial resistance of these microorganisms is essential for the clinician to ensure a correct diagnosis and timely, effective therapy.

## Figures and Tables

**Figure 1 antibiotics-12-01222-f001:**
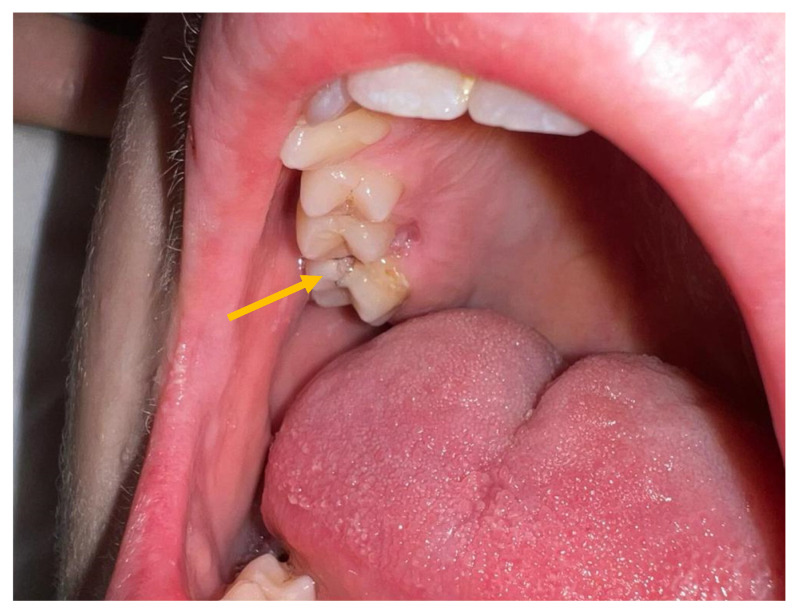
Oral cavity of the patient at the time of admission. Presence of destructive caries in the upper dental arch.

**Figure 2 antibiotics-12-01222-f002:**
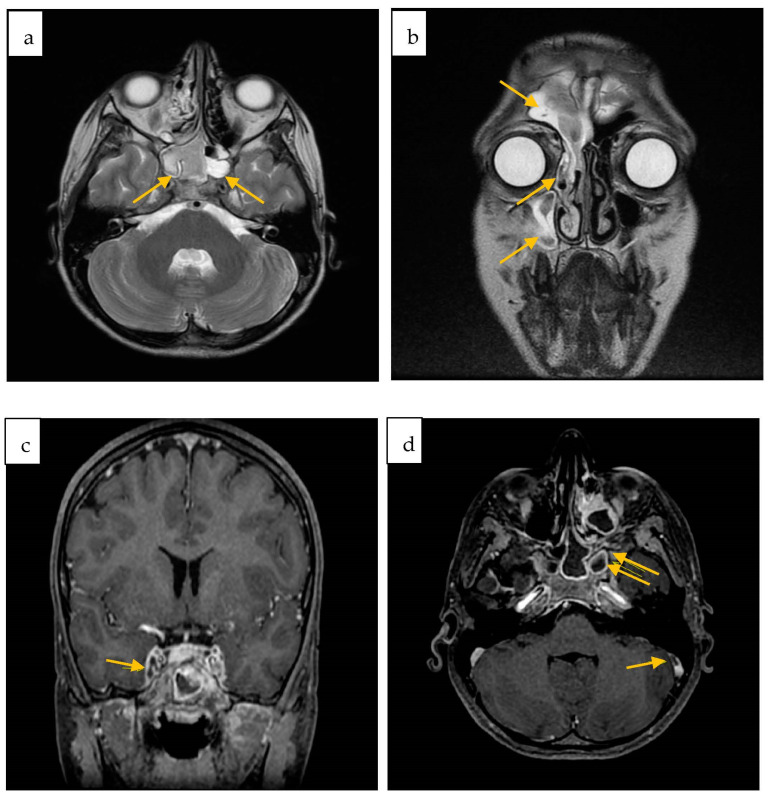
Contrast-enhanced brain MRI axial (**a**) and coronal (**b**) FSE T2w sinusitis of both sphenoid sinuses (**a**), right maxillary and frontal sinus, and ipsilateral ethmoidal cells (**b**); (**c**) Inhomogeneous opacification of the left cavernous sinus; (**d**) Defect of opacification of the right sigmoid sinus (arrow) as for partial thrombosis. Right sphenoid sinus and maxillary sinus sinusitis (double arrow).

**Figure 3 antibiotics-12-01222-f003:**
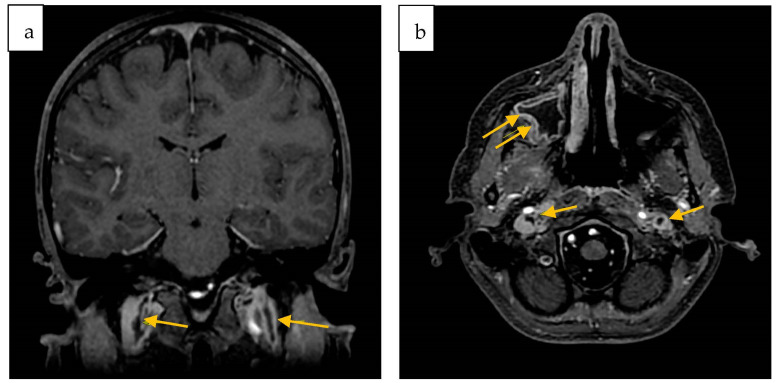
Second contrast-enhanced brain MRI. Coronal (**a**) and Axial (**b**) 3D FSPGR defect of opacification of both jugular veins (arrows); (**b**) right maxillary sinusitis (double arrows).

**Figure 4 antibiotics-12-01222-f004:**
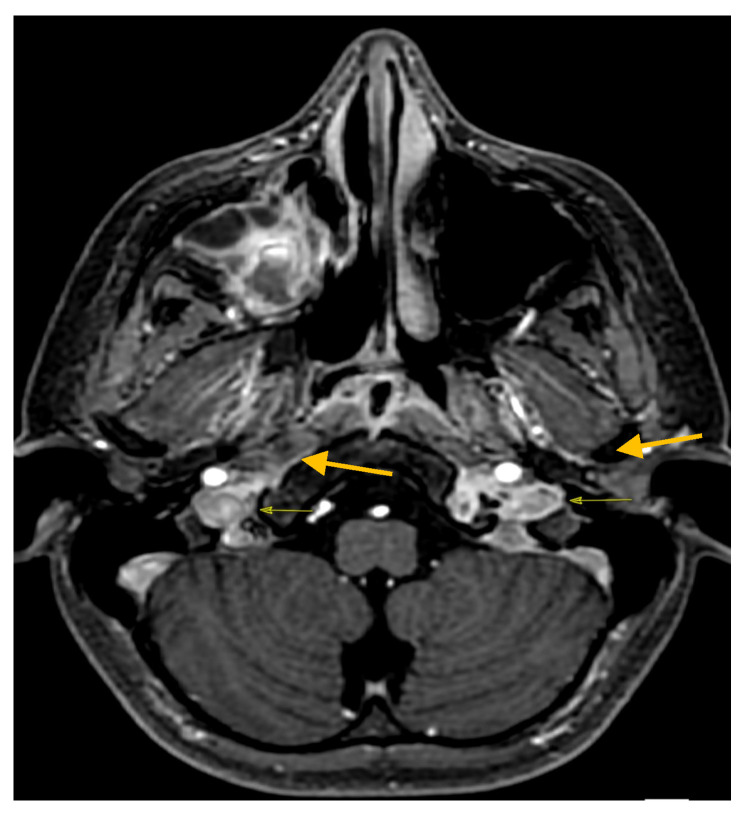
Control MRI post discharge axial 3D FSPGR resolution of jugular vein thrombosis.

**Figure 5 antibiotics-12-01222-f005:**
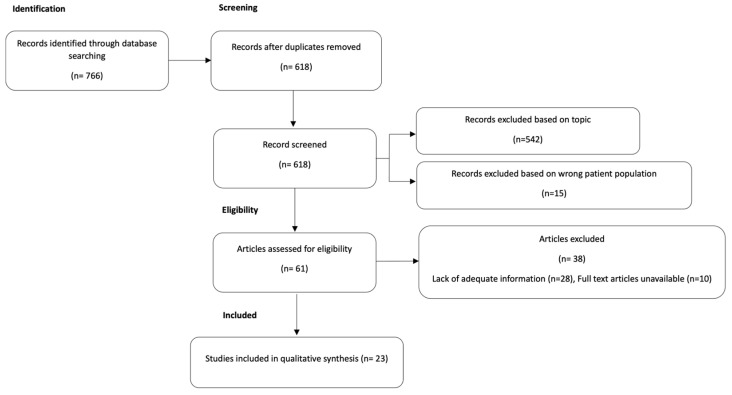
PRISMA study flow diagram: flow diagram of study identification, screening, eligibility, and included studies.

**Figure 6 antibiotics-12-01222-f006:**
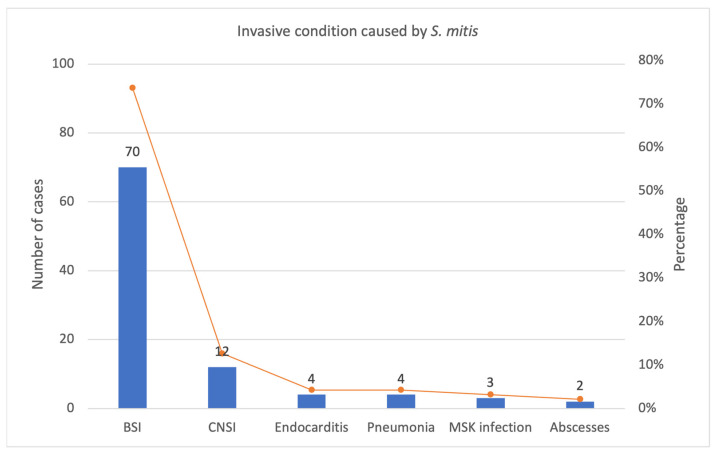
Most frequent invasive condition caused by *S. mitis* in our review: number of cases and percentage. BSI: bloodstream infections, CNSI: central nervous system infections, MSK: musculoskeletal.

**Table 1 antibiotics-12-01222-t001:** Laboratory findings.

	Results	Measuring Unit	Reference Values
WBC	12.8	10^3^/uL	4.00–10.00
Neutrophils	11.6	10^3^/uL	2.70–6.70
Lymphocytes	0.50	10^3^/uL	1.00–2.20
RBC	4.1	10^6^/uL	3.74–4.92
Hemoglobin	10.8	g/dL	11.0–14.3
Platelet count	255	10^3^/uL	180.0–299.0
C-reactive protein	16.86	mg/dL	<0.5

**Table 2 antibiotics-12-01222-t002:** Reported cases of invasive disease caused by *S. mitis*.

Author/Country/ Year [Ref.]	Age	Sex	Pre-Existing Disease	Risk Factors	*S. mitis* Isolation Sample	Medical Condition	Therapy	Outcome
Goldfarb et al./ USA/1984 [24]	9 d	M	Hydrocephalus	ventriculoperitoneal shunt	CSF	meningitis	nafcillin + ampicillin + gentamicin → vancomycin	healed
	2 y	F	AML	CVC	CSF	meningitis	Penicillin + gentamicin → ampicillin	healed
Hellwege et al./ Germany/1984 [25]	2 d	F	No	No	CSF	meningitis	ampicillin + gentamicin → penicillina G	healed
Bignardi et al./ UK/1989 [16]	2 d	F	No	No	CSF	meningitis	penicillin G + netilmicin → penicillin G	healed
Tobias et al./ USA/1991 [17]	5 y	M	AML	CVC	blood culture	sepsis	vancomycin + amikacin + ticarcillin	healed
	6 y	M	AML	CVC	blood culture	sepsis	vancomycin + amikacin + ceftazidime	healed
	10 y	F	AML	CVC	blood culture	sepsis	vancomycin + amikacin + ceftazidime	healed
	1 y	F	AML	CVC	blood culture	sepsis	vancomycin + ceftazidime	healed
Balkundi et al./ USA/1997 [6]	7 y	F	Burkitt lymphoma	CVC	NR	meningitis	tobramycin + nafcillin + ticarcillin → vancomycin	healed
	6 y	F	ALL	CVC	NR	meningitis	vancomycin + ceftazidime → intraventricular vancomycin	died
	9 y	M	ALL	CVC	NR	meningitis	vancomycin + ceftazidime → vancomycin	healed
Rieske et al./ Germania/1997 [15]	15 y	M	ALL	CVC	blood culture	bacteremia	erythromycin → ampicillin	healed
	8 y	F	ALL	CVC	blood culture	pneumonia	cefotaxime + gentamicin → azlocillin	healed
	6 m	M	AML	CVC	blood culture	pneumonia	imipenem + vancomycin	healed
	4 y	M	AML	CVC	blood culture	septic shock	cefotaxime + gentamicin + vancomycin	died
	6 y	F	AML	CVC	blood culture	otomastoiditis	ceftazidime + gentamicin + ampicillin	healed
	10 y	M	AML	CVC	blood culture	pneumonia	azlocillin + ceftazidime + gentamicin	healed
	15 y	M	AML	CVC	blood culture	sepsis	piperacillina + ceftazidime + gentamicina	died
	10 y	F	AML	CVC	blood culture	NR	ceftazidime + gentamicin → piperacillin	healed
	6 m	M	Osteopetrosis	CVC	blood culture	bacteremia	NR	healed
Legendre et al./ France/2000 [23]	1 y	F	No	ventricular septal defect	NR	endocarditis	NR	healed
Jaing et al./ Japan/2002 [5]	6 y	M	ANLL	CVC	blood culture and CSF	meningitis	vancomycin + ceftriaxone	healed
Ahmed et al./ UK/2003 [11]	6.6 y *	13 M, 9 F	Oncological disease	CVC	blood culture	sepsis	NR	21 healed 1 died
Kennedy et al./ USA/2004 [21]	8 y	F	No	ventricular septal defect	blood culture	endocarditis	vancomycin + ceftriaxone + gentamicin	healed
Taketani et al./ Japan/2009 [20]	2 y	M	No	No	blood culture	endocarditis	ampicillin/sulbactam + meropenem → vancomycin + gentamicin + rifampicin → linezolid	healed
Nomura et al./ Finland/2011 [10]	7 y	F	No	No	blood culture	osteomyelitis	clindamycin	healed
Yiş R et al./ Turkey/2011 [2]	8y	F	No	No	CSF	meningitis	ceftriaxone	healed
Yiş U et al./ Turkey/2012 [22]	6 y	M	No	No	CSF	sinusitis meningitis	ceftriaxone + vancomycin	healed
Vazquez Melendez et al./ USA/2014 [19]	2 y	F	No	No	pleural fluid	pneumonia	clindamycin + ceftriaxone → amoxicillin/clavulanate	healed
Nielsen et al./ UK/2015 [18]	9 y *	28 NR	hematological diseases	CVC	blood culture	sepsis	NR	28 healed
	2 y	NR	Medulloblastoma	CVC	blood culture	septic shock	ceftazidime + amikacin → teicoplanin	healed
	11 y	NR	ALL	CVC	blood culture	septic shock	ceftazidime + amikacin + teicoplanin	healed
	17 y	NR	ALL	CVC	blood culture	septic shock	piperacillin/tazobactam + gentamicin → teicoplanin	died
	16 y	NR	aplastic anemia	CVC	blood culture	septic shock	piperacillin/tazobactam + teicoplanin	died
	18 y	NR	ALL	CVC	blood culture	septic shock	piperacillin/tazobactam + gentamicin → teicoplanin	died
	7 y	NR	ALL	CVC	blood culture	septic shock	piperacillin/tazobactam + gentamicin → teicoplanin	healed
Esposito et al./ Italy/2015 [26]	9.5 y *	5 NR	heart disease	heart disease	blood culture	endocarditis	NR	healed
Buldu et al./ UK/2016 [12]	4 y	M	No	No	blood culture	pyomyositis of the hip	benzylpenicillin → phenoxymethylpenicillin	healed
Basaranoglu et al./ Japan/2018 [8]	9 y	M	AML	CVC	blood culture	bacteremia	NR	healed
	2 y	M	AML	CVC	blood culture	bacteremia	NR	healed
	13 y	M	AML	CVC	blood culture	bacteremia	NR	healed
	6 m	M	Osteopetrosis	CVC	blood culture	bacteremia	NR	healed
Imhof et al./ Germany/2018 [13]	5 y	F	metastatic nephroblastoma	CVC	biopsy sample	liver and lung abscess	cefuroxime	healed
Watanabe et al./ Japan/2019 [14]	7 m	F	No	No	drainage material	retropharyngeal abscess	meropenem	healed
Blázquez-Gamero et al./ Spain/2021 [9]	3 m	M	Swachman-Diamond Syndrome, COVID-19	No	blood culture	bacteremia	NR	healed
Fukayama et al./ Japan/2021 [4]	14 y	F	Gorham-Stout syndrome	skull base osteolysis and CSF leak	CSF	meningitis	penicillin G	healed
Colomba C. et al./Italy/2023	12 y	M	No	No	sinus drainage material	sinusitis, meningitis, CSVT	ceftriaxone + vancomycin + metronidazole → cefotaxime	healed

Y years; m months; d days; y * mean age; F female; M male; ALL acute lymphoid leukemia; AML acute myeloid leukemia; ANLL acute nonlymphocytic leukemia; CVC central venous catheters; CSF cerebrospinal fluid; CSVT cerebral sino-venous thrombosis; NR not reported.

## Data Availability

Data is contained within the article.

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
