# Peer review of "Streptococcus mitis as a New Emerging Pathogen in Pediatric Age: Case Report and Systematic Review"

_antibiotics, 2023, doi:10.3390/antibiotics12071222_

Round 1
Reviewer 1 Report
This article investigated a critical issue: the invasive infections caused by Streptococcus mitis in immunocompromised and immunocompetent pediatric patients. However, the manuscript has several issues that need to be addressed before considering it for publication:
Line 2: The name of the bacteria should be written in the full-spelled-out format (Streptococcus mitis) in the title and not in the short format (S. mitis).
Line 15: The abbreviation “VGS” stands for “Viridans Group Streptococci,” not “Viridans Streptococci Group.”
Lines 29 – 37: These three paragraphs have the same general idea. It is recommended to combine them into a single paragraph.
Lines 38 – 44: These two paragraphs can be combined into a single paragraph.
Lines 286 – 297: These four paragraphs have the same general idea. It is recommended to combine them into a single paragraph.
Lines 324 – 335: The information here appears redundant since they have already been mentioned between lines 312 – 323.
Lines 339 – 340: Reference [2] is not consistent with the other references (the study title is written between brackets, and there is no DOI mentioned).
Lines 348 – 349: Reference [6] is miswritten since there are unrecognizable symbols in the study title.
Lines 352 – 353: Reference [8] is inconsistent with the other references since the author’s name was written in uppercase letters.
Lines 376 – 378: Reference [17] is inconsistent with the other references since the author’s name was written in uppercase letters.
Lines 379 – 380: Reference [18] is inconsistent with the other references since all the authors’ names were written in uppercase letters.
Lines 393 – 396: References [24] and [25] are inconsistent with the other references since the authors’ names were written as initials, with the last names not fully recognizable. Also, there is no DOI mentioned for these two references.
Lines 399 – 400: Reference [27] is inconsistent with the other references since the study title was written in uppercase letters.
Lines 466 – 467: Reference [56] is inconsistent with the other references since the author’s name was written in uppercase letters.
Lines 486 – 487: Reference [65] is redundant since it has already been mentioned as reference [61]. This must be fixed in the reference list and the in-text citation.
The manuscript needs to be revised for some grammatical errors, improper use of punctuation, and difficult-to-read sentence structures.
Author Response
Please see the attachment. All edits in the manuscript are highlighted in track change function.

Reviewer 2 Report
The article entitled ”S. mitis as a New Emerging Pathogen in Pediatric Age: 2 Case Report and Systematic Review” describes and interesting case report and provides a comprehensive review of the existing data in the literature concerning this elusive pathogen. It is a well written article and using PRISMA and PROSPERO provides great credibility. I have some comments that I will present as follows:
· Line 35-37 provide a reference.
· Figure 1-2-3 should be bigger and maybe add some arrows where the main findings are presented.
· The CT-scan images are available? If yes, please add them too.
· Line 57-66 I would advise to create a table where you add the patient’s laboratory findings and also the normal values in your laboratory, so it is easy to see the changes that were present for this patient.
· Line 67-69 why use antibiotics in a culture negative bacterial meningitis? You mean an empirical treatment was used (I suppose so since you mention it in line 274)? Please explain and rephrase.
· Line 89 – why did you perform echocardiography and color Doppler (with capital D)? Please explain/rephrase/erase.
· Line 100 – pansinusitis usually implies the involvement of all of the sinuses. Please change to sinusitis since it is obvious what you mean.
· Line 113 - that ”it also revealed” you mean the MRI? If yes, add that section to the above paragraph.
· Line 117 – Do you have the control MRI images? If yes, please add them too.
· Line 129 – What do you mean by life threatening disease? Please explain and give examples of what you included.
· Line 140 – you mentioned 100 patients and after 101 – please adjust the information.
· Figure 4 – you mentioned that some articles were excluded using aggregate data only – do you mean the lack of adequate information (as stated in line 144)? If yes, please keep the same phrasing since it is better than ”aggregate data only”.
· In Table 1 some data are not clear - Ahmed et al – oncological disease (what type?), Nielsen et al – hematological disease (what type?) and Esposito et al (heart disease - what type?) -Please write the exact diagnosis.
· Figure 5 – in the annotations you have CSI and in the image you have CNSI – please change so it is clear.
· Line 197 – you mentioned that in 2 cases the outcome was not available. Why did you chose to include them? The outcome of the infections was one of the inclusion criteria, right? Please explain, rephrase or erase those cases.
· Line 210-217 – The close genetic correlation does not necessarily mean that virulence is the same. The are several morphological and pathogenetic differences between Streptococcus pneumoniae and Streptococcus mitis. Please add a paragraph where you detail the pathogenetic factors described for Streptococcus mitis and perhaps explain if they are present or not in Streptococcus pneumoniae.
· Line 299 – Needs to be aligned.
· References should be uniformly cited.
Author Response
Please see the attachment. Please note all changes to the manuscript are available in Track Change function.
